# Effect of Digital Intervention on Nurses’ Knowledge About Diabetic Foot Ulcer: A Quasi-Experimental Study

**DOI:** 10.3390/ijerph22111610

**Published:** 2025-10-22

**Authors:** Kauan Gustavo de Carvalho, Lídya Tolstenko Nogueira, Daniel de Macêdo Rocha, Jefferson Abraão Caetano Lira, Álvaro Sepúlveda Carvalho Rocha, Sandra Marina Gonçalves Bezerra, Luciana Tolstenko Nogueira, Claudia Daniella Avelino Vasconcelos, Iara Barbosa Ramos, Laelson Rochelle Milanês Sousa

**Affiliations:** 1Department of Nursing, Federal University of Piauí, Teresina 64049-550, Brazil; kauancarvalho2008@gmail.com (K.G.d.C.); lidyatn@gmail.com (L.T.N.); j.abraaolira@gmail.com (J.A.C.L.); .; cdavb2010@gmail.com (C.D.A.V.); 2Department of Nursing, Federal University of Mato Grosso do Sul, Coxim 79400-000, Brazil; iara.ramos@ufms.br; 3Department of Nursing, State University of Piauí, Teresina 64049-550, Brazil; sandramarina20@hotmail.com; 4Department of Medicine, State University of Piauí, Teresina 64049-550, Brazil; lutolstenko@hotmail.com; 5Nursing Course, State University of Maranhão, Coroatá 65665-000, Brazil; laelsonmilanes@gmail.com

**Keywords:** diabetic foot, nurse, knowledge, mainstreaming, education, primary health care

## Abstract

Educational strategies based on technological models that integrate the dimensions of prevention, screening, and treatment of diabetic foot ulcers are emerging as promising methods to improve nurses’ knowledge, skills, and clinical competencies in primary care. In this investigation, we evaluated the effectiveness of a digital education program, mediated by a virtual learning environment, in enhancing nurses’ clinical knowledge about diabetic foot ulcers. This quasi-experimental intervention study was conducted with 114 nurses, selected for convenience, from the five health districts that make up primary care in the municipality of Teresina, Brazil. Two stages, separated by the educational intervention, allowed us to measure their knowledge levels before and after the implementation of the digital technology. A characterization form and the Nurse Knowledge Assessment Questionnaire on Diabetic Foot were used to evaluate the outcomes. The McNemar test compared the pre- and post-intervention knowledge levels, while accuracy rate-based parameters allowed for the classification of results into performance categories. The intervention effect size was estimated using Cohen’s d test. Results showed substantial improvements in knowledge, particularly in domains related to definition (*p* = 0.002), risk factors (*p* < 0.001), associated complications (*p* < 0.001), signs and symptoms of neuropathies (*p* < 0.001), application of tests to assess protective sensation (*p* < 0.001) and foot biomechanics (*p* < 0.001), risk classification (*p* < 0.001), and prevention strategies (*p* < 0.001), with performance ratings predominantly “good” or “excellent” after the intervention. The effect size for paired samples was large (Cohen’s dz = 1.82), based on the total knowledge scores. Findings support the effectiveness signal of the virtual learning environment for knowledge improvement; however, without a control group, we cannot rule out testing effects. Controlled or stepped-wedge trials should confirm causality.

## 1. Introduction

Diabetic foot ulcer represents a significant chronic complication of diabetes mellitus (DM) and is responsible for substantial rates of morbidity, mortality, disability, and amputations worldwide [1]. Considered a complex condition, this neuropathic, ischemic, or neuro-ischemic lesion is generally located in the lower limbs, has a multifactorial origin, and affects people of all age groups and genders, particularly those with poor glycemic control and long disease duration [2].

Prevalence and recurrence of diabetic foot ulcers remain high globally, with substantial morbidity and costs; early risk stratification in primary health care (PHC) is a recognized gap, particularly in nurse-led practice. The effects of this scenario include increased costs related to hospitalizations and adjuvant treatments, loss of productivity, social isolation, psychological distress, and reduced quality of life for the affected population [3].

Global projections characterize diabetic foot ulcer as a major public health problem. The annual incidence ranges from 19% to 34%, and recurrence may occur in up to 65% of cases [4]. Brazil has one of the highest prevalence rates in the world and remains among the countries with the largest absolute number of cases. Although public prevention policies are in place, initiatives for early recognition in PHC, risk classification, and appropriate management are limited, incipient, and often neglected [5,6]. Added to this are limitations in access to specialized services and gaps in nurses’ knowledge, which pose a challenge to ensuring prompt and effective responses for lesion prevention in the Brazilian population [7].

Knowledge gaps among nurses regarding the clinical management of diabetic foot ulcers are frequently reported in primary care and hinder the effectiveness of population-based screening and prevention programs [7]. In Brazil, clinical protocols establish the use of the 10 g monofilament test, the 128 Hz tuning fork, the Achilles reflex, and dorsalis pedis and posterior tibial pulses to classify risk based on the assessment of plantar protective and vibratory sensitivity as well as neural function and peripheral perfusion. Failures in this process affect timely diagnosis, promote lesion worsening, increase the incidence of infections, and contribute to the recurrence of ulcers and preventable amputations [8,9].

In this context, educational strategies based on technological models stand out as promising methods to improve nurses’ knowledge, skills, and clinical competencies in the management of diabetic foot ulcers [10,11]. In the literature, the virtual learning environment (VLE) is widely recognized as a digital tool with potential for the creation and sharing of content in interactive formats, respecting individual learning paces [12]. Its design can involve personalized multimedia resources, simulated practical activities, discussion forums, and formative assessments, and has shown positive impacts on clinical performance, decision-making, and adherence to evidence-based protocols [13,14].

Although digital educational interventions aimed at the care of diabetic foot ulcers have been developed in recent years, few studies in Brazil have evaluated the VLE as a methodology for the continuing education of nurses in the management of diabetic foot ulcers. Thus, there is a clear knowledge gap regarding the existence of educational programs that integrate, in an articulated manner, the dimensions of prevention, screening, and treatment, especially directed toward primary health care (PHC) nurses [15,16]. Studies that use the VLE as a tool for continuing education are promising and show potential to optimize work processes, improve professional performance, reduce unfavorable health indicators, and mitigate outcomes that may impact the population’s quality of life.

Given the magnitude of the problem and the need for educational interventions to ensure qualified, resolutive, and evidence-based care, this study analyzed the effectiveness of a digital education program mediated by the virtual learning environment in improving nurses’ clinical knowledge about diabetic foot ulcers. The hypothesis adopted was that the educational intervention would promote a significant improvement in nurses’ knowledge, strengthening their role in PHC.

## 2. Materials and Methods

### 2.1. Study Design

This was a single-group pre–post design without a concurrent control; therefore, the estimates may be influenced by testing effects and unmeasured confounders.

### 2.2. Participants

We approached 124 nurses: 120 consented and 114 completed both assessments and were analyzed. The selection was made for convenience in the five health districts that make up the PHC of the municipality of Teresina, Piauí, Brazil. The sample size was calculated using the proportional technique for finite populations considering a 95% confidence level, 5% margin of error, and an assumed prevalence of diabetic foot ulcers of 50% [17].

The study design and sampling techniques were established to meet the practical characteristics of the research, particularly regarding participant availability and the need to measure individual changes in knowledge before and after the intervention. Despite the absence of a control group, validated instruments, robust tests, and standardized intervention procedures were employed to reduce the risk of bias.

Nurses with a minimum of six months’ experience in primary care and no prior exposure to educational practices related to the management of diabetic foot ulcers were included. The minimum time of practice was defined to ensure that the participants had consistent practical experience in the study context as well as consider that this period was essential for nurses to understand the epidemiological profile of the population served, establish bonds with users, and develop confidence in clinical assessment and care interventions.

Exclusion criteria included professionals on leave from work, those who had undergone training on diabetic foot ulcers during the study period, and those who did not complete all study phases in full. In this study, the inclusion of participants on leave or recently trained could have introduced bias related to the absence of care practice or the influence of recent updates on their level of knowledge. Participants who did not fully complete the data collection instruments were also excluded to ensure internal validity, consistency, and reliability of the results as well as comparability of the outcomes before and after the intervention.

### 2.3. Intervention

The intervention consisted of an educational approach mediated by a VLE, developed on the Moodle platform, and hosted at https://pediabetico.net/. The content covered topics related to prevention, risk classification, assessment, and the management of diabetic foot ulcers, targeted at nurses working in primary care.

The program was delivered on two separate days to accommodate all participants. Initially, an introductory guide and a tutorial with platform usage recommendations were provided, followed by video lectures, multimedia case studies, discussion forums, and quizzes for learning assessment and reinforcement. After the intervention presentation, the nurses had unrestricted access to the VLE for 30 days to explore the content and materials provided.

The VLE was developed in a previous study following a systematic review [18] and validated by field experts and IT professionals. This digital resource consists of five educational modules that structured the intervention: (1) Epidemiological aspects of diabetic foot, foot anatomy and biomechanics; (2) Pathophysiology of diabetic foot; (3) Clinical examination of the feet in people with diabetes mellitus and risk classification for ulceration; (4) Health education for foot self-care in people with diabetes mellitus and appropriate footwear; (5) Treatment of diabetic foot. Preliminary tests confirmed the tool’s validity and usability as well as its potential for integration into continuing education practices in PHC [19].

Each module was composed of theoretical and practical content presented through illustrative video lessons lasting between 10 and 50 min. In addition, the modules provided supplementary materials, interactive quizzes (such as questions on the formation of plantar arches and their biomechanical function), discussion forums, and study slides. Interaction was monitored by the researchers and field specialists, and was encouraged during discussion forums and through feedback on quizzes.

### 2.4. Study Procedures

The operational phases occurred from September to December 2023, beginning with contact with the Teresina Municipal Health Foundation to present the project, identify, select, and recruit participants. Given the relevance of the study and its potential healthcare impacts, this institution—responsible for coordinating the municipal health network—authorized the full release of all PHC nurses to participate in the study.

The study was conducted in two stages, separated by the educational intervention. The pre-test phase collected participant demographic data and assessed the baseline knowledge levels. Following the intervention, a post-test was conducted after 30 days to reassess knowledge. This time frame was chosen to align with previous studies and to allow for delayed evaluation, thereby measuring knowledge retention over time [20,21].

Participant characterization was performed using a validated and adapted questionnaire comprising three dimensions: sociodemographic, occupational, and educational aspects. Variables included sex, age, self-reported race/ethnicity, family income, marital status, number of employment relationships, academic qualifications, years of experience, work hours, participation in scientific events, and availability of instruments for diabetic foot ulcer risk assessment.

Knowledge assessment before and after the intervention was performed using the Nurse Knowledge Investigation Questionnaire on Diabetic Foot (QICEPeD), developed and validated by Félix in 2021 [22]. This questionnaire consists of eighteen items distributed across sixteen domains and three dimensions: two theoretical and one practical.

Theoretical Dimension I includes six domains and six items assessing knowledge on the definition of diabetic foot, risk factors, complications, prevention, and signs and symptoms of motor and autonomic neuropathy. Theoretical Dimension II comprises six domains addressing tests, frequency of application, examination sites, and the interpretation of results for protective sensation and foot biomechanics assessments. The practical dimension includes four domains and six items evaluating foot examination practices by nurses, common problems, risk classification systems, and barriers to performing foot examinations in PHC [22].

All domains were assessed through multiple-choice questions. No cutoff point was established for the raw score. Therefore, the scoring limit was not fixed, and the classification of demonstrated knowledge was based on the analysis of the percentage of correct answers and the proportion of correct responses in each assessed domain.

### 2.5. Data Analysis and Processing

Data were double entered into Excel 2013 spreadsheets and analyzed using the Statistical Package for Social Sciences (SPSS) version 22.0. Sociodemographic, occupational, and educational characteristics were analyzed through descriptive statistics including mean, standard deviation, minimum, maximum, absolute frequency, and relative frequency.

In the pre-test phase, baseline knowledge was assessed based on the percentage of correct answers, and the calculated values reflected the number of correct responses obtained by participants across the different dimensions and domains covered by the instrument. The primary outcome of this study was the total knowledge score.

Knowledge assessment employed the McNemar test, which compared the percentages of correct answers before and after the intervention. This is a widely referenced non-parametric statistical technique used to evaluate changes in dichotomous categorical variables when the same participants are assessed at two distinct time points. In before-and-after quasi-experimental studies, the McNemar test allows for the verification of whether there is a significant difference between the proportions of correct and incorrect answers pre- and post-intervention. Thus, it becomes an appropriate, precise, and consistent analytical tool for validating results in nursing education studies when the goal is to measure changes in performance, knowledge, behaviors, and clinical practices within the same group.

Effect size was estimated using Cohen’s d for paired samples considering a 95% confidence interval. This estimate was defined by the following formula:dm = (M1 − M2)/((DP1 + DP2)/2)
where M represents the mean and SD the standard deviation [23,24]. The effect size was classified as small (>0.2), medium (>0.5), or large (>0.8). Abdullah’s parameters were used to classify the level of knowledge into performance categories according to the percentage of correct answers: poor (<55%), fair (55–70%), good (70–85%), and excellent (>85%) [25].

### 2.6. Ethical Considerations

The study was approved by the Research Ethics Committee of the Federal University of Piauí, under opinion number 5.179.989. Participation was voluntary and contingent upon signing an informed consent form.

## 3. Results

### 3.1. Participant Characteristics

This study included 114 nurses, the majority of whom were female (93.9%), with a mean age of 44.54 (SD = 10.38) years, a family income between BRL 7100 and BRL 22,000 (63.2%), and self-identified as mixed race (“pardo”) (67.5%). Specialization was the predominant highest academic qualification (64.0%), with 5.3% holding a specialization in wound care. Most participants had not attended scientific events in the past three years (58.8%) and had no publications in the study field (66.7%).

The occupational profile revealed a long length of service in the Family Health Strategy (FHS), with 90.4% having worked in this role for more than 10 years. Maintaining two employment contracts (59.6%) and working between 45 and 60 h per week (47.4%) were common conditions among participants.

Regarding foot assessments in individuals with diabetes mellitus, 60.5% reported rarely performing them during nursing consultations, and 67.5% did not use specific instruments. When instruments were available, the combination of monofilament, tuning fork, and neurological hammer was the most frequently used method (67.5%). A considerable proportion of participants (59.6%) also reported limitations in the availability of material resources in their health services (Table 1).

### 3.2. Assessment of Nurses’ Knowledge on the Management of Diabetic Foot Ulcers

Table 2 describes the knowledge demonstrated by nurses when assessing Theoretical Dimension I of the QICEPeD as well as presenting a comparison of the percentage of correct answers and performance before and after the educational intervention. The highest accuracy rates were concentrated in the domains “Prevention of Foot Ulcers” (99.1%; 100%), “Risk Factors for Diabetic Foot Syndrome” (74.4%; 93.9%), and “Definition of Diabetic Foot” (79.8%; 91.2%), which showed significant improvements after the VLE-mediated intervention and were classified as excellent.

The lowest knowledge scores before the VLE-mediated educational intervention involved complications (50%) and the signs and symptoms of autonomic neuropathy (46.5%). In these domains, the proportion of correct answers increased significantly after the educational approach, resulting in marked improvements, with scores rising to 79.8% and 76.3%, respectively, indicating a good level of demonstrated knowledge (*p* < 0.001). Similarly, recognition of motor neuropathy improved from 60.5% (“fair”) to 78.1% (“good”), also with statistical significance (*p* < 0.001). It should be noted that the item-level results are exploratory, and interpretation should focus on the domain-level and total scores.

In Theoretical Dimension II, the results also showed a marked increase in the percentage of correct answers among nurses after the educational intervention, with significant improvements in knowledge across all evaluated domains and items (*p* < 0.05). Before the intervention, low performance was observed regarding the application and interpretation of specific tests to assess protective sensation, particularly in the use of the tuning fork test (62.3%), pinprick test (64.0%), and Achilles tendon reflex hammer test (44.7%).

Similarly, the domains assessing knowledge of the frequency of foot assessments, the recommended number of test applications, and biomechanical assessment also had limited percentages of correct answers and showed significant gains after the intervention, moving from poor or regular classifications to good or excellent. Significant increases in knowledge were also observed in other evaluated domains: “Sites for Application of the 10 g Semmes-Weinstein Monofilament Test” (*p* < 0.001), “Recommended Number of Test Applications” (*p* < 0.001), “Interpretation of Tests for Assessing Loss of Protective Sensation” (*p* < 0.001), “Biomechanical Foot Assessment” (*p* = 0.013), and “Frequency of Foot Assessment According to Risk Classification” (*p* = 0.001) (Table 3).

Regarding practical knowledge, the nurses’ clinical performance in foot assessment for people with DM was unsatisfactory, with most domains classified as “poor” or “regular”. Improvements were observed in the ability to identify changes and complications such as recognizing problems related to self-care (86.0% to 94.7%; *p* = 0.021), use of inappropriate footwear (83.3% to 93.0%; *p* = 0.027), and decreased protective sensation (58.8% to 80.7%; *p* < 0.001). Significant increases in correct answers also occurred in identifying more severe clinical conditions, such as foot ulcers (24.6% to 36.0%; *p* < 0.001) and a history of amputations (19.3% to 31.5%; *p* < 0.001), suggesting greater knowledge of diagnostic strategies and heightened sensitivity for recognizing risk signs.

Regarding the frequency of risk assessment, all categories showed statistically significant increases (*p* < 0.001). As for the factors hindering foot examination, the professionals’ perception also evolved, with increased recognition of lack of training (67.5% to 86.8%; *p* < 0.001), work overload (*p* = 0.031), and lack of adequate instruments (48.2% to 72.8%; *p* < 0.001) as relevant barriers to practical performance. The domains related to foot examination (47.4%) and shoe inspection (48.2%) remained classified as “poor” even after the intervention, although a slight percentage increase in knowledge was observed (Table 4).

The effect size analysis showed a Cohen’s d = 1.820. This coefficient substantially exceeded the conventional threshold for a large effect (d > 0.8) and indicates an expressive, relevant, and consistent impact of the educational intervention on the level of knowledge of PHC nurses regarding the management of diabetic foot ulcers.

## 4. Discussion

Managing risk, implementing preventive measures, and ensuring the proper management of diabetic foot ulcers are the responsibilities of nurses in primary care and require specific knowledge, clinical competencies, and practical skills conducive to safe, sustainable, effective, and evidence-based care [26,27]. Recognizing the potential of educational technologies as valid, reliable, and effective strategies is a promising approach to identify theoretical and practical gaps, clinical inconsistencies, and unmet needs as well as propose interventions capable of overcoming care challenges and generating effective responses for ulcer prevention in the Brazilian population.

This study was aimed at analyzing the effects of a digital intervention on nurses’ knowledge regarding diabetic foot ulcers. The technological foundation was based on the virtual learning environment (VLE), and the effect analyses highlighted its potential, validity, and effectiveness in ensuring positive impacts on the studied outcome, suggesting that this technology represents an important method for continuing education and nursing training.

Substantial improvements in knowledge were observed after the educational intervention, especially in the domains involving definitions, risk factors, associated complications, signs and symptoms of neuropathies, application of tests for assessing sensation and foot biomechanics, risk classification, and prevention strategies, which were rated as good or excellent after the intervention. These significant advances have also been reported in other studies, which identified improvements of up to 22.49% in the level of knowledge presented by nurses following a digital intervention [28,29,30].

The enhanced knowledge levels can be explained by a combination of factors including the flexibility and interactivity of the VLE. This technology allows participants to study at their own pace and integrates active methodologies such as instructional video lessons, multimedia resources, case studies, and discussion forums that stimulate critical reflection and reinforce learning retention. In this way, the educational intervention promoted meaningful learning, enhancing the nurses’ qualification for diabetic foot management in primary health care.

In this study, the parameters proposed by Abdullah were used to classify the nurses’ performance into categories. This methodology has established scientific relevance, allows for the objective and replicable comparison of results in different contexts, enhances analytical capacity, and guides decision-making and the formulation of institutional policies for nursing team training and capacity building [25].

### Practical Implications for Continuing Education Policies

In Brazil, PHC represents an important component of the Unified Health System (SUS), aligned with public policies on humanization, patient safety, and continuing education [31]. At this level of care, nurses act as strategic agents, and their in-depth knowledge is essential to promote user autonomy and self-care as well as ensure the prevention, risk classification, and proper management of diabetic foot ulcers [32].

In this regard, understanding the clinical definition and factors associated with diabetic foot ulcers, domains with excellent knowledge levels after the educational intervention, is essential for nursing practice and involves identifying neuropathic, vascular, and infectious changes as well as recognizing determinants for risk stratification. Systematically addressing these conceptual foundations is crucial for the early identification of structural, biomechanical, and functional changes that may increase the vulnerability and estimated risk in this population [33].

Similarly, having adequate knowledge to recognize complications and the signs and symptoms of motor and autonomic neuropathies enables nurses to act before irreversible lesions develop, intensifying preventive measures in the presence of trophic changes that predispose to cracks, fissures, or infections. Targeted interventions are essential to restore functionality and quality of life for affected individuals as well as reduce mortality and the economic costs to the health system [34].

The impact of early identification, risk recognition, and targeted interventions for modifiable risk factors is well-documented in the literature and strengthens prevention efforts, particularly in vulnerable contexts. Moreover, it allows for care planning aimed at minimizing the occurrence of more severe outcomes [35]. Nurses in this study demonstrated considerable knowledge in this evaluated domain, showing both theoretical and practical foundations to implement care before critical events occur, particularly infections, recurrences, and amputations.

Although the intervention improved knowledge about the application of tests to assess the loss of protective sensation in at-risk feet, this skill remains a persistent challenge in nursing practice. The identified limitations revealed a prior lack of knowledge about the correct anatomical points for test application, the recommended testing frequency, and the appropriate interpretation of results, indicating gaps that compromise the effectiveness of screening and follow-up of at-risk patients.

In clinical practice, incorrect or irregular performance of tests as well as imprecise interpretation, results in significant failures in the identification of peripheral neuropathies, proper risk classification, and the timely implementation of necessary interventions [36,37]. In this study, this domain revealed substantial knowledge gaps prior to the intervention, and even with improvements, this area of care should remain a focus of ongoing training. Similarly, knowledge regarding interventions, foot problems, footwear inspection, recognition of autonomic neuropathy, risk classification, and factors that hinder foot examination was low, remaining limited even after the educational intervention, highlighting a structural challenge in the nursing field within primary care.

These results may be associated with the practical and clinical nature of these competencies, which require not only the acquisition of theoretical knowledge, but also the development of technical skills, detailed observational ability, and interpretation of signs. Given this, it is relevant to consider the adoption of complementary educational strategies to enhance the effectiveness of the teaching-learning process.

This gap has been pointed out in different studies, which have shown that a significant number of nurses are unfamiliar with the protocols and clinical guidelines, either due to gaps in initial training or the absence of continuing education in care services [38,39]. Structural limitations, scarce material resources, and work overload may also explain the impacts on nurses’ ability to adapt their approach to ensure adequate assessment.

In this research, persistent knowledge gaps were also observed after the intervention. Such results suggest that educational approaches, even when well-structured, may not be sufficient to produce lasting improvements in theoretical knowledge and clinical practice [40]. Nursing practice could benefit from the incorporation of other teaching strategies into ongoing training programs.

Resources such as clinical simulation, supervised practical workshops, and active methodologies based on real-life scenarios can promote the consolidation of more complex skills by allowing nurses to experience situations close to everyday practice. The use of technological resources, such as augmented reality and virtual reality, also emerges as a promising alternative, providing immersion and experimentation in simulated contexts. Thus, the integration of virtual learning with supervised practice can be a strategic approach to address specific gaps and enhance professional qualification in the management of diabetic foot [40].

Therefore, the VLE-mediated educational strategy proved to be an effective, accessible tool aligned with the routines of PHC nursing professionals, capable of reducing geographical and financial barriers as well as expanding access to and the reach of training initiatives. Furthermore, this study demonstrates the potential of digital education for integration into continuing health education policies, impacting care quality, the prevention of chronic conditions, and the promotion of self-care across different contexts and levels of care.

This study was conducted following a robust methodology, and while providing strong evidence, it is important to acknowledge its limitations. Although the assessed knowledge encompassed theoretical and practical dimensions as well as primary care, this investigation did not measure the clinical impacts in real-world settings or across the different scenarios that comprise Brazil’s healthcare network. The absence of a control group, convenience sampling, the potential influence of self-reported practices, the follow-up period, and the conduct of the study in a single municipality represent important limitations regarding causal inference and generalization of the results, which should be addressed in future research. Additionally, outcomes related to satisfaction and self-confidence with the educational intervention were not evaluated.

Acknowledging these gaps allows for the design of future plans to expand the representativeness of the target population and guide longitudinal studies that evaluate the effectiveness of this educational strategy on the clinical risk assessment rates, prevention indicators, and the management of diabetic foot ulcers and their associated complications.

## 5. Conclusions

This study demonstrated the effects of virtual learning environment-based education on nurses’ knowledge regarding the management of diabetic foot ulcers. By using a validated digital technology, improved performance was observed in both theoretical and practical knowledge related to prevention, recognition of risk factors, and the use, frequency, and interpretation of methods for risk assessment. Advances were also noted in knowledge about complications, conditions suggestive of greater clinical severity, implementation of diagnostic strategies, and sensitivity in identifying risk signs. Recognition of complications and symptoms of autonomic neuropathy as well as footwear inspection remained challenging areas.

The effectiveness of the educational strategy was confirmed by the learning effect size, which showed a substantial, relevant, and consistent impact on the assessed outcome. Although the results are promising, the effects on clinical practice and patient outcomes were not evaluated and remain a priority for future studies.

## Figures and Tables

**Table 1 ijerph-22-01610-t001:** Sociodemographic and occupational characteristics of nurses working in the Family Health Strategy.

Category	N	%	M (SD)	Max	Min
Age			44.54 (10.38)	73	22
Sex					
Female	107	93.9			
Male	7	6.1			
Race/Color					
White	28	24.6			
Black	9	7.9			
Mixed race (“pardo”)	77	67.5			
Family income					
<BRL 2900	3	2.6			
BRL 2900–7100	37	32.5			
BRL 7100–22,000	72	63.2			
>BRL 22,000	2	1.8			
Marital status					
Without partner	38	33.3			
With partner	76	66.7			
Number of employment contracts					
1	42	36.8			
2	68	59.6			
≥3	4	3.5			
Specialization in wound care					
No	108	94.7			
Yes	6	5.3			
Length of service					
1–5 years	7	6.1			
6–10 years	4	3.5			
>10 years	103	90.4			
Weekly working hours					
30–44 h	46	40.4			
45–60 h	54	47.4			
>60 h	14	12.3			
Highest qualification					
Bachelor’s degree	15	13.2			
Specialization	73	64.0			
Master’s degree	21	18.4			
Doctorate	5	4.4			
Participation in scientific events (last 3 years)					
No	67	58.8			
Yes	47	41.2			
Foot assessment during nursing consultation					
No	27	23.7			
Yes, rarely	69	60.5			
Yes, frequently	18	15.8			
Instruments used for foot assessment					
No instruments used	69	60.5			
10 g monofilament	18	15.8			
Monofilament + tuning fork + neurological hammer	77	67.5			
Monofilament + tuning fork	34	29.8			
Service provides instruments					
No	68	59.6			
Yes, some instruments	45	39.5			
Yes, all instruments	1	0.9			

**Table 2 ijerph-22-01610-t002:** Nurses’ knowledge in Theoretical Dimension I of the QICEPeD before and after the educational intervention (N = 114).

Questions	Before			After			*p*-Value
	N	%	Classification	N	%	Classification	
Definition of diabetic foot	91	79.8	Good	104	91.2	Excellent	0.002
Risk factors	86	75.4	Good	107	93.9	Excellent	<0.001
Complications	57	50.0	Poor	91	79.8	Good	<0.001
Signs and symptoms of motor neuropathy	69	60.5	Regular	89	78.1	Good	<0.001
Signs and symptoms of autonomic neuropathy	53	46.5	Poor	87	76.3	Good	<0.001
Prevention of foot ulcers	113	99.1	Excellent	114	100.0	Excellent	1.000

**Table 3 ijerph-22-01610-t003:** Nurses’ knowledge in Theoretical Dimension II of the QICEPeD before and after the educational intervention (N = 114).

Questions	Before			After			*p*-Value
	N	%	Classification	N	%	Classification	
**Tests to assess loss of protective sensation in at-risk feet**							
10 g monofilament test	98	86.0	Excellent	109	95.6	Excellent	0.003
128 Hz tuning fork test	71	62.3	Regular	105	92.1	Excellent	<0.001
Pinprick sensation test	73	64.0	Regular	101	88.6	Excellent	<0.001
Achilles reflex hammer test	51	44.7	Poor	86	74.6	Good	<0.001
Sites for application of the 10 g Semmes-Weinstein monofilament test							
Hallux	59	51.8	Poor	105	92.1	Excellent	<0.001
1st metatarsal	48	42.1	Poor	98	86.0	Excellent	<0.001
3rd metatarsal	46	40.4	Poor	96	84.2	Good	<0.001
5th metatarsal	49	43.0	Poor	93	81.6	Excellent	<0.001
Recommended number of test applications	30	26.3	Poor	100	87.7	Excellent	<0.001
Interpretation of tests for assessing loss of protective sensation	78	68.4	Regular	102	89.5	Excellent	<0.001
Biomechanical assessment of the feet	97	85.1	Good	108	94.7	Excellent	0.013
Frequency of foot assessment according to risk classification	36	34.2	Poor	85	74.6	Good	<0.001

**Table 4 ijerph-22-01610-t004:** Nurses’ practical knowledge in the QICEPeD before and after the educational intervention (N = 114).

Description	Before			After			*p*-Value
	N	%	Classification	N	%	Classification	
Intervention							
Examines feet of people with DM	54	47.4	Poor	58	50.9	Poor	0.125
Examines shoes of people with DM	55	48.2	Poor	60	52.6	Poor	0.125
Foot problems							
Lack of knowledge about self-care	98	86.0	Excellent	108	94.7	Excellent	0.021
Use of inadequate footwear	95	83.3	Good	106	93.0	Excellent	0.027
Inadequate foot hygiene	100	87.7	Excellent	104	91.2	Excellent	0.503
Cracks	94	82.5	Good	101	88.6	Excellent	0.248
Absence of pulses	28	24.6	Poor	31	27.2	Poor	0.755
Decreased sensation	66	58.8	Regular	92	80.7	Good	<0.001
Burning pain	57	50.0	Poor	64	56.5	Regular	0.311
Calluses	93	81.6	Good	99	86.8	Excellent	0.238
Deformities	39	34.2	Poor	44	38.6	Poor	0.511
Gait alteration	48	42.1	Poor	56	49.1	Poor	0.268
Ingrown nail	53	46.5	Poor	60	52.6	Poor	0.337
Fungal infection	62	52.6	Poor	67	58.8	Regular	0.265
Bacterial infection	56	49.1	Poor	65	57.0	Regular	0.150
External trauma	54	47.4	Poor	64	56.1	Regular	0.110
Gangrene	27	23.7	Poor	29	25.4	Poor	0.500
Foot ulcer	28	24.6	Poor	41	36.0	Poor	<0.001
Amputation	22	19.3	Poor	36	31.5	Poor	<0.001
Frequency							
People without loss of sensation and without PAD	25	21.9	Poor	96	84.2	Good	<0.001
With loss of sensation without deformities	23	20.2	Poor	98	86.0	Excellent	<0.001
With loss of sensation with deformities	36	31.1	Poor	84	73.7	Good	<0.001
With PAD and loss of sensation	32	28.1	Poor	79	69.3	Regular	<0.001
With PAD and no loss of sensation	32	28.1	Poor	83	72.4	Good	<0.001
With history of ulcer	18	15.8	Poor	69	60.5	Regular	<0.001
Factors hindering foot examination							
Lack of training	77	67.5	Regular	99	86.8	Excellent	<0.001
Lack of instruments	55	48.2	Poor	83	72.8	Good	<0.001
Work overload	60	52.6	Poor	66	57.9	Regular	0.031
Low patient adherence	44	38.9	Poor	52	45.6	Poor	0.008

**Legend:** DM—diabetes mellitus; PAD—peripheral arterial disease.

## Data Availability

The original contributions presented in this study are included in the article. Further inquiries can be directed to the corresponding author.

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
