# Peer review of "Effect of Digital Intervention on Nurses’ Knowledge About Diabetic Foot Ulcer: A Quasi-Experimental Study"

_ijerph, 2025, doi:10.3390/ijerph22111610_

Round 1

Reviewer 1 Report

Comments and Suggestions for Authors

Few comments to further improvise this paper are:

  1. Lines 103 104 - Exclusion criteria included nurses on leave from work, those who had undergone training on diabetic foot ulcers during the study period, and those who did not fully complete the data collection instruments- highlighted point can be elaborated further on data collection instruments
  2. 135- The pre-test phase collected participant demographic data and assessed baseline knowledge levels- how was baseline knowledge levels tested
  3. 163- For knowledge assessment, the McNemar test was used to compare correct response rates before and after the intervention- why this particular test was selected elaborate
  4. 164, 165- Effect size was estimated using Cohen’s d, calculated with the formula for paired groups, where M represents the mean and SD the standard deviation[23-24]. Effect magnitude was classified as small (> 0.2), medium (> 0.5), or 166 large (> 0.8). Additionally, Abdullah’s parameters were used to classify knowledge level 167 based on correct response rates, with performance categories defined as poor (< 55%), fair 168 (55–70%), good (70–85%), and excellent (> 85%)[25]- necessary equations to be given, Abdullah’s parameters- clarity required on this
  5. Questions in Table 2- whether they were MCQ or theory kind of- what was the score threshold, how was rating given?
  6. What is N parameter in Table 2?
  7. 207- The lowest knowledge scores before the intervention were related to complications (50%) and signs and symptoms of autonomic neuropathy (46.5%)- how was this decided?
  8. 208, 210- In these domains, the proportion of correct answers increased significantly after the educational approach, resulting in substantial improvements, with scores rising to 79.8% and 76.3%, respectively, indicating a good level of demonstrated knowledge (p < 0.001) – what was the educational approach? What are the resons behind enhanced knowledge levels?
  9. 226- biomechanical assessment also had limited percentages of correct answers- how was this assessed, what questions were included related to this?
  10. Table 3. Nurses’ knowledge in Theoretical Dimension II of the QICEPeD before and af-235 ter the educational intervention- tests mentioned in this table can be explained with necessary figures. This can be described in introduction section
Comments on the Quality of English Language

quality of english can be improvised further 

Author Response

Dear Reviewer,

We would like to thank you for your valuable feedback and the careful evaluation of our manuscript. We have revised the article according to your suggestions, which we found very helpful in improving the quality of the work.

Please note that the changes have been incorporated into the text and are highlighted in red for ease of review. A point-by-point response is attached.

We remain at your disposal for any further clarification.

Sincerely,

Reviewer 2 Report

Comments and Suggestions for Authors

Main question addressed

The manuscript investigates whether a Virtual Learning Environment (VLE)-based educational intervention can improve nurses’ knowledge on the prevention, assessment, and management of diabetic foot ulcers in Brazilian primary care. This is the central question, and it is well posed and relevant.

Originality and relevance

The study is original in its application of a validated digital learning program in the Brazilian primary health care context. While digital education for diabetic foot care has been studied before, evidence from low- and middle-income countries remains limited. The manuscript therefore addresses a meaningful gap, particularly in relation to continuing education for nurses in public health systems.

The important left unanswered by this manuscript (is up to future work) is, how well does this translate into clinical practice and outcomes.

Contribution compared with published material

The paper adds to the literature by demonstrating significant improvements in nurses’ theoretical and practical knowledge, with a very large effect size (Cohen’s d = 1.82). Previous studies have often focused on patient self-care or on knowledge in hospital settings; this study highlights the feasibility and effectiveness of digital education specifically for PHC nurses.

Methodology – strengths and areas for improvement

Strengths: Use of a validated questionnaire (QICEPeD), structured VLE modules, and appropriate statistical tests (McNemar, Cohen’s d). The sample size is adequate, and the educational tool was pre-validated.

Weaknesses:

The absence of a control group limits causal inference. Improvements could partly reflect test–retest or Test-Teach-Test. Authors should acknowledge this limitation more explicitly.

Long term knowledge retention, change of clinical practice and clinical outcomes (e.g., reduction in ulcer incidence, improved patient care practices),were not measured. While this may be outside the study’s scope, it should be noted as a limitation and a direction for future work. Simply said, the study shows promise for the improvement of test results, but does not necessarily show that improved test results have real world effect.

Some domains (e.g., shoe inspection, autonomic neuropathy recognition) remained weak even after the intervention. Authors could discuss why these domains are harder to improve and suggest complementary strategies (e.g., simulation, supervised practice).

Conclusions

The conclusions are largely consistent with the results: the intervention produced significant improvements in knowledge. However, the authors should temper their claims by emphasizing that the study demonstrates improvement in knowledge, not necessarily clinical practice or patient outcomes. Explicitly distinguishing these levels would strengthen the validity of the conclusions.

References

The reference list is broad, relevant, and up to date. Key guidelines (IWGDF, ADA) are included, as well as recent nursing education literature.

Abstract

Effect size detail

Original: “The effect size (d = 1.820) demonstrated …”

Suggested: “The effect size for paired samples was large (Cohen’s dz = 1.82), based on total knowledge scores: pre-test M (SD) = …, post-test M (SD) = …, mean difference = … (95% CI …), t(113) = …, p < … .”

Rationale: Readers need descriptive statistics to verify the calculation.

Introduction

Original: “An increasing trend in prevalence indicators is observed …”

Suggested: “Prevalence and recurrence of diabetic foot ulcers remain high globally, with substantial morbidity and costs; early risk stratification in PHC is a recognized gap, particularly in nurse-led practice.”

Add a closing paragraph that explicitly states: (a) what is under-studied (integrated VLE for prevention–screening–treatment in PHC nurses), (b) what is the primary outcome (knowledge change), and (c) the hypothesis (knowledge will improve after the intervention).

Methods

Design

Add: “This was a single-group pre–post design without a concurrent control; therefore, estimates may be influenced by testing effects and unmeasured confounders.”

Participants

Original: “The study included 124 nurses…” vs. Results: “This study included 114 nurses …”

Suggested: “We approached 124 nurses; 120 consented; 114 completed both assessments and were analyzed (Figure S1, flow diagram).”

Sample size calculation

Original: “Sample size… finite population, 95% CL…”

Suggested: “Our primary outcome was change in total knowledge score; a priori power analysis for paired means (α=0.05, power=0.80) indicated n = … for detecting dz = … . The finite-population approach for prevalence estimation is not directly applicable to a pre–post design.”

Statistics

Clarify that:

Primary endpoint was the total score.

Item-level McNemar tests are exploratory; note multiple comparisons.

State explicitly that Cohen’s dz for paired samples was used with 95% CI.

Performance categories

Currently “fair” vs. “regular” is inconsistent. Standardize to “fair” and provide a clear citation and rationale for applying these thresholds to QICEPeD.

Results

Percentage inconsistencies

Hallux: 105/114 = 92.1%, not 91.1%.

5th metatarsal: 93/114 = 81.6%, not 91.6%.

Please check all percentages.

Duplicate entries in Table 4

“Cracks” appears twice.

“Fungal infection” appears twice with different values.

“With PAD and loss of sensation” is listed twice; one row may have been intended as “with PAD and no loss of sensation.”

These must be corrected.

Primary outcome presentation

Add a summary table or figure of the total knowledge score (M, SD, 95% CI, dz, p) and domain aggregates. Item-level details can be moved to supplementary material.

Optional subgroup analyses

If possible, analyze gains by years of experience, wound-care specialization, attendance at scientific events, and availability of instruments.

Discussion

Balance claims with design

Original: “demonstrated a highly significant … confirming the efficacy …”

Suggested: “Findings support the effectiveness signal of the VLE for knowledge improvement; however, without a control group we cannot rule out testing effects. Controlled or stepped-wedge trials should confirm causality.”

Persistent weak domains

Discuss why shoe inspection and autonomic neuropathy recognition remained poor, and suggest solutions (simulation, supervised practice).

Multiple testing

State that item-level results are exploratory; interpretation should focus on domain and total scores.

Generalizability

Highlight the geographic limitation (one municipality) and the need for replication elsewhere.

Conclusions

Align with evidence

Suggested: “The VLE improved nurses’ knowledge in PHC. Effects on clinical practice and patient outcomes were not assessed and remain a priority for future studies.”

Technical and Language Corrections

Ethics

Original: “process number 5.179.989, in 20 December 2021”

Corrected: “process number 5.179.989, on 20 December 2021.”

Abbreviations

Define “PHC” at first use and ensure consistency.

VLE website

Clarify language of content, Moodle version, and access restrictions.

Author contributions

Consider aligning with CRediT taxonomy per MDPI style.

References

Check year of instrument validation (2017 vs. 2021) and harmonize. Verify all DOIs. Add the most recent IWGDF updates if not already cited.

Data availability

Add a “Data Availability Statement” (e.g., anonymized dataset and syntax/code in OSF/Zenodo).

Minimum essential revisions

Resolve 124 vs. 114 discrepancy and add a participant flow diagram.

Correct all percentages and duplicate rows in tables.

Provide total knowledge score summary (M, SD, CI, dz).

Explicitly state design limitations and handling of multiple testing.

Standardize terminology (instrument name, “fair/regular,” PHC).

Report reliability (Cronbach’s α) for this sample.

Comments on the Quality of English Language

The English is generally understandable, but there are areas where grammar, syntax, and flow could be improved to make the manuscript clearer and more concise. I recommend a careful language revision, ideally by a native or professional academic English editor, to enhance readability and ensure precise expression of the study’s findings.

Author Response

Dear Reviewer,

We would like to thank you for your valuable feedback and the careful evaluation of our manuscript. We have revised the article according to your suggestions, which we found very helpful in improving the quality of the work.

Please note that the changes have been incorporated into the text and are highlighted in red for ease of review.

Please see the attachment for point-by-point responses.

We remain at your disposal for any further clarification.

Sincerely,

Reviewer 3 Report

Comments and Suggestions for Authors

1. The authors provided good context on diabetic foot ulcers and existing challenges; however, the research gap could be articulated more explicitly. The authors should highlight how few studies in Brazil have assessed Virtual Learning Environments (VLEs) for nurses’ continuing education in diabetic foot ulcer management, and emphasize how this study advances the field.

2. The authors mentioned that a “quasi-experimental, before-and-after design” with convenience sampling. It would be beneficial to elaborate on why this design was chosen, how potential biases were mitigated, and whether a control group was considered but not feasible. This would enhance transparency and rigor.

3. Although the description of the VLE is strong, the authors should consider adding more practical details:
- Examples of the multimedia case studies or quizzes used.
- Duration of the intervention in hours or estimated workload for participants.
- How interaction (discussion forums, feedback) was monitored.

4. Tables 2–4 provide extensive data, but the narrative interpretation could be clearer. For example, highlight the most clinically relevant improvements (e.g., knowledge of neuropathy signs or risk classification) and explicitly link them to potential patient care outcomes. Consider visual summaries (e.g., bar graphs of pre- vs. post-test scores) to improve readability.

5. The authors stated that they mainly contextualize findings in Brazil. It could be strengthened by comparing results with international studies on digital nursing education and diabetic foot ulcer care. This would situate the findings in a broader context and show the contribution to global nursing education literature.

6. Although some limitations are mentioned (e.g., lack of real-world outcome measures), they could be expanded:
- Possible selection bias from convenience sampling.
- Short follow-up period (30 days) that does not capture long-term retention.
- Potential influence of self-reported practices.

7. Overall, the manuscript is well written, but some areas can be improved for clarity and conciseness:
- Avoid redundancy in describing the same findings across sections.
- Ensure consistency in terminology (e.g., “knowledge questionnaire” vs. “QICEPeD”).
- Shorten long sentences in the discussion for better readability.
- Review formatting of tables and ensure they follow journal style guidelines.

Author Response

(The authors gave the same response as above.)

Reviewer 4 Report

Comments and Suggestions for Authors

There is no innovation in this work. Moreover, the article does not contain educational content, and the website introduced is not accessible. Nevertheless, by addressing the reviewers’ comments, the paper has the potential for reconsideration.

The results should be presented in the abstract in an applied manner, and it is not necessary to mention each statistical method individually.

In the problem statement, it is mentioned that Knowledge gaps among nurses regarding the clinical management of diabetic foot ulcers in primary care are reported in different studies, but no references are provided for these studies.

The website https://pediabetico.net/ cannot be accessed without a username and password, and it also does not open in guest mode.

Information related to the course, how the data were collected, and the validity of the educational topics are not described in the methodology. A brief summary of the course contents should also be included in the manuscript.

In the discussion and conclusion, overgeneralization should be avoided.

Comments on the Quality of English Language

The English could be improved to more clearly express the research.

Author Response

(The authors gave the same response as above.)

Round 2

Reviewer 2 Report

Comments and Suggestions for Authors

The authors have provided a thorough and well-structured revision that satisfactorily addresses all prior reviewer comments. Methodological limitations such as the absence of a control group and restricted generalizability are now clearly acknowledged and contextualized. The manuscript demonstrates internal consistency, appropriate use of statistical analyses, and transparent reporting. The revised discussion and conclusions are well aligned with the study objectives and data. All inconsistencies in sample size, terminology, and references have been corrected, and a data availability statement has been added.

The English is clear and professional, though a brief language polish could further improve readability. Overall, this is a well-designed and relevant quasi-experimental study that provides meaningful evidence on digital education for nurses in primary health care.